# LANGUAGE MODELS REPRESENT SPACE AND TIME

**Wes Gurnee & Max Tegmark**
Massachusetts Institute of Technology
{wesg, tegmark}@mit.edu

## ABSTRACT

The capabilities of large language models (LLMs) have sparked debate over whether such systems just learn an enormous collection of superficial statistics or a set of more coherent and grounded representations that reflect the real world. We find evidence for the latter by analyzing the learned representations of three spatial datasets (world, US, NYC places) and three temporal datasets (historical figures, artworks, news headlines) in the Llama-2 family of models. We discover that LLMs learn *linear* representations of space and time across multiple scales. These representations are robust to prompting variations and unified across different entity types (e.g. cities and landmarks). In addition, we identify individual "space neurons" and "time neurons" that reliably encode spatial and temporal coordinates. While further investigation is needed, our results suggest modern LLMs learn rich spatiotemporal representations of the real world and possess basic ingredients of a world model.

## 1 INTRODUCTION

Despite being trained to just predict the next token, modern large language models (LLMs) have demonstrated an impressive set of capabilities (Bubeck et al., 2023; Wei et al., 2022), raising questions and concerns about what such models have actually learned. One hypothesis is that LLMs learn a massive collection of correlations but lack any coherent model or "understanding" of the underlying data generating process given text-only training (Bender & Koller, 2020; Bisk et al., 2020). An alternative hypothesis is that LLMs, in the course of compressing the data, learn more compact, coherent, and interpretable models of the generative process underlying the training data, i.e., a *world model*. For instance, Li et al. (2022) have shown that transformers trained with next token prediction to play the board game Othello learn explicit representations of the game state, with Nanda et al. (2023) subsequently showing these representations are linear. Others have shown that LLMs track boolean states of subjects within the context (Li et al., 2021) and have representations that reflect perceptual and conceptual structure in spatial and color domains (Patel & Pavlick, 2021; Abdou et al., 2021). Better understanding of if and how LLMs model the world is critical for reasoning about the robustness, fairness, and safety of current and future AI systems (Bender et al., 2021; Weidinger et al., 2022; Bommasani et al., 2021; Hendrycks et al., 2023; Ngo et al., 2023).

In this work, we take the question of whether LLMs form world (and temporal) models as literally as possible—we attempt to extract an actual map of the world! While such spatiotemporal representations do not constitute a dynamic causal world model in their own right, having coherent multi-scale representations of space and time are basic ingredients required in a more comprehensive model.

Specifically, we construct six datasets containing the names of places or events with corresponding space or time coordinates that span multiple spatiotemporal scales: locations within the whole world, the United States, and New York City in addition to the death year of historical figures from the past 3000 years, the release date of art and entertainment from 1950s onward, and the publication date of news headlines from 2010 to 2020. Using the Llama-2 (Touvron et al., 2023) and Pythia Biderman et al. (2023) family of models, we train linear regression probes (Alain & Bengio, 2016; Belinkov, 2022) on the internal activations of the names of these places and events at each layer to predict their real-world location (i.e., latitude/longitude) or time (numeric timestamp).

These probing experiments reveal evidence that models build spatial and temporal representations throughout the early layers before plateauing at around the model halfway point with larger models

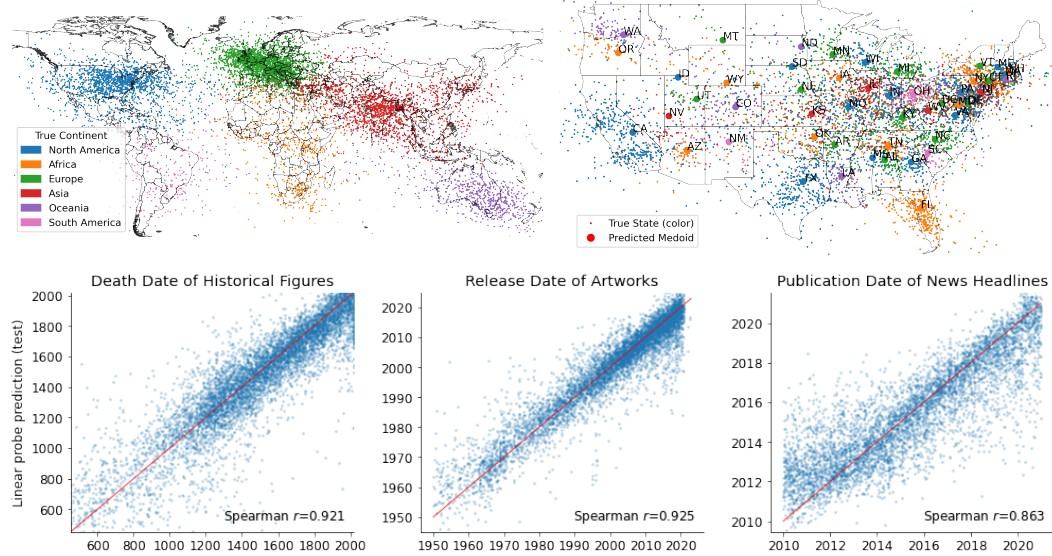

Figure 1: Spatial and temporal world models of Llama-2-70b. Each point corresponds to the layer 50 activations of the last token of a place (top) or event (bottom) projected on to a learned linear probe direction. All points depicted are from the test set.

consistently outperforming smaller ones (§ 3.1). We then show these representations are (1) linear, given that nonlinear probes do not perform better (§ 3.2), (2) fairly robust to changes in prompting (§ 3.3), and (3) unified across different kinds of entities (e.g. cities and natural landmarks). We then conduct a series of robustness checks to understand how our probes generalize across different data distributions (§ 4.1) and how probes trained on the PCA components perform (§ 4.2). Finally, we use our probes to find individual neurons which activate as a function of space or time and use basic causal interventions to verify their importance in spatiotemporal modeling, providing strong evidence that the model is truly using these features (§ 5).

## 2 EMPIRICAL OVERVIEW

### 2.1 SPACE AND TIME RAW DATASETS

To enable our investigation, we construct six datasets of names of *entities* (people, places, events, etc.) with their respective location or occurrence in time, each at a different order of magnitude of scale. For each dataset, we included multiple types of entities, e.g., both populated places like cities and natural landmarks like lakes, to study how unified representations are across different object types. Furthermore, we maintain or enrich relevant metadata to enable analyzing the data with more detailed breakdowns, identify sources of train-test leakage, and support future work on factual recall within LLMs. We also attempt to deduplicate and filter out obscure or otherwise noisy data.

**Space** We constructed three datasets of place names within the world, the United States, and New York City. Our world dataset is built from raw data queried from DBpedia Lehmann et al. (2015). In particular, we query for populated places, natural places, and structures (e.g. buildings or infrastructure). We then match these against Wikipedia articles, and filter out entities which do not have at least 5,000 page views over a three year period. Our United States dataset is constructed from DB-Pedia and a census data aggregator, and includes the names of cities, counties, zipcodes, colleges, natural places, and structures where sparsely populated or viewed locations were similarly filtered out. Finally, our New York City dataset is adapted from the NYC OpenData points of interest dataset (NYC OpenData, 2023) containing locations such as schools, churches, transportation facilities, and public housing within the city.

**Time** Our three temporal datasets consist of (1) the names and occupations of historical figures who died between 1000BC and 2000AD adapted from (Annamoradnejad & Annamoradnejad, 2022); (2) the titles and creators of songs, movies, and books from 1950 to 2020 constructed from DBpedia with the Wikipedia page views filtering technique; and (3) New York Times news headlines from 2010-2020 from news desks that write about current events, adapted from (Bandy, 2021).

Table 1: Entity count and representative examples for each of our datasets.

| Dataset | Count | Examples |
|---------|-------|----------|
| World | 39585 | "Los Angeles", "St. Peter's Basilica", "Caspian Sea", "Canary Islands" |
| USA | 29997 | "Fenway Park", "Columbia University", "Riverside County" |
| NYC | 19838 | "Borden Avenue Bridge", "Trump International Hotel" |
| Figures | 37539 | "Cleopatra", "Dante Alighieri", "Carl Sagan", "Blanche of Castile" |
| Art | 31321 | "Stephen King's It", "Queen's Bohemian Rhapsody" |
| Headlines | 28389 | "Pilgrims, Fewer and Socially Distanced, Arrive in Mecca for Annual Hajj" |

## 2.2 MODELS AND METHODS

**Data Preparation** All of our experiments are run with the base Llama-2 (Touvron et al., 2023) series of auto-regressive transformer language models, spanning 7 billion to 70 billion parameters. For each dataset, we run every entity name through the model, potentially prepended with a short prompt, and save the activations of the hidden state (residual stream) on the last entity token for each layer. For a set of $n$ entities, this yields an $n \times d_{model}$ activation dataset for each layer.

**Probing** To find evidence of spatial and temporal representations in LLMs, we use the standard technique of probing Alain & Bengio (2016); Belinkov (2022), which fits a simple model on the network activations to predict some target label associated with labeled input data. In particular, given an activation dataset $\boldsymbol{A} \in \mathbb{R}^{n \times d_{model}}$, and a target $\boldsymbol{Y}$ containing either the time or two-dimensional latitude and longitude coordinates, we fit linear ridge regression probes

$$\hat{\boldsymbol{W}} = \underset{\boldsymbol{W}}{\arg\min} \|\boldsymbol{Y} - \boldsymbol{A}\boldsymbol{W}\|_2^2 + \lambda\|\boldsymbol{W}\|_2^2 = (\boldsymbol{A}^T\boldsymbol{A} + \lambda\boldsymbol{I})^{-1}\boldsymbol{A}^T\boldsymbol{Y}$$

yielding a linear predictor $\hat{\boldsymbol{Y}} = \boldsymbol{A}\hat{\boldsymbol{W}}$. High predictive performance on out-of-sample data indicates that the base model has temporal and spatial information linearly decodable in its representations, although this does not imply that the model actually uses these representations (Ravichander et al., 2020). In all experiments, we tune $\lambda$ using efficient leave-out-out cross validation (Hastie et al., 2009) on the probe training set.

## 2.3 EVALUATION

To evaluate the performance of our probes we report standard regression metrics such as $R^2$ and Spearman rank correlation on our test data (correlations averaged over latitude and longitude for spatial features). An additional metric we compute is the *proximity error* for each prediction, defined as the fraction of entities predicted to be closer to the target point than the prediction of the target entity. The intuition is that for spatial data, absolute error metrics can be misleading (a 500km error for a city on the East Coast of the United States is far more significant than a 500km error in Siberia), so when analyzing errors per prediction, we often report this metric to account for the local differences in desired precision.

## 3 LINEAR MODELS OF SPACE AND TIME

### 3.1 EXISTENCE

We first investigate the following empirical questions: do models represent time and space at all? If so, where internally in the model? Does the representation quality change substantially with model scale? In our first experiment, we train probes for every layer of Llama-2-{7B, 13B, 70B} and

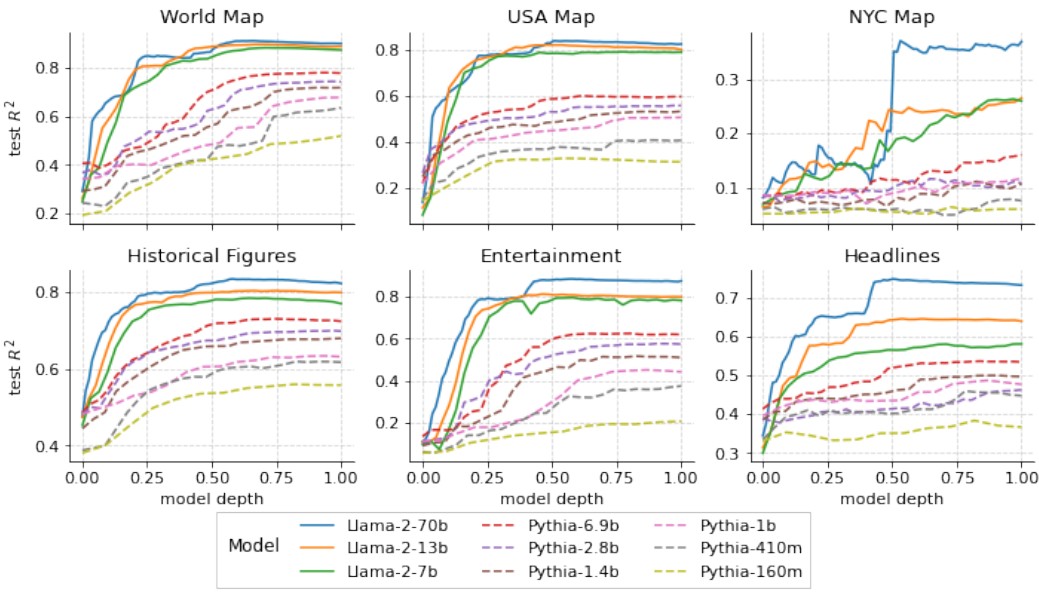

Figure 2: Out-of-sample $R^2$ for linear probes trained on every model, dataset, and layer.

Pythia-{160M, 410M, 1B, 1.4B, 2.8B, 6.9B} for each of our space and time datasets. Our main results, depicted in Figure 2, show fairly consistent patterns across datasets. In particular, both spatial and temporal features can be recovered with a linear probe, these representations smoothly increase in quality throughout the first half of the layers of the model before reaching a plateau, and the representations are more accurate with increasing model scale. The gap between the Llama and Pythia models is especially striking, and we suspect is due to the large difference in pre-training corpus size (2T and 300B tokens respectively). For this reason, we report the rest of our results on just the Llama models.

The dataset with the worst performance is the New York City dataset. This was expected given the relative obscurity of most of the entities compared with other datasets. However, this is also the dataset where the largest model has the best relative performance, suggesting that sufficiently large LLMs could eventually form detailed spatial models of individual cities.

## 3.2 LINEAR REPRESENTATIONS

Within the interpretability literature, there is a growing body of evidence supporting the *linear representation hypothesis* that features within neural networks are represented linearly, that is, the presence or strength of a feature can be read out by projecting the relevant activation on to some feature vector (Mikolov et al., 2013b; Olah et al., 2020; Elhage et al., 2022b). However, these results are almost always for binary or categorical features, unlike the continuous features of space or time.

To test whether spatial and temporal features are represented linearly, we compare the performance of our linear ridge regression probes with that of substantially more expressive nonlinear MLP probes of the form $W_2\text{ReLU}(W_1 x + b_1) + b_2$ with 256 neurons. Table 2 reports our results and shows that using nonlinear probes results in minimal improvement to $R^2$ for any dataset or model. We take this as strong evidence that space and time are also represented linearly (or at the very least are linearly decodable), despite being continuous.

## 3.3 SENSITIVITY TO PROMPTING

Another natural question is if these spatial or temporal features are sensitive to prompting, that is, can the context induce or suppress the recall of these facts? Intuitively, for any entity token, an autoregressive model is incentivized to produce a representation suitable for addressing any future possible context or question.

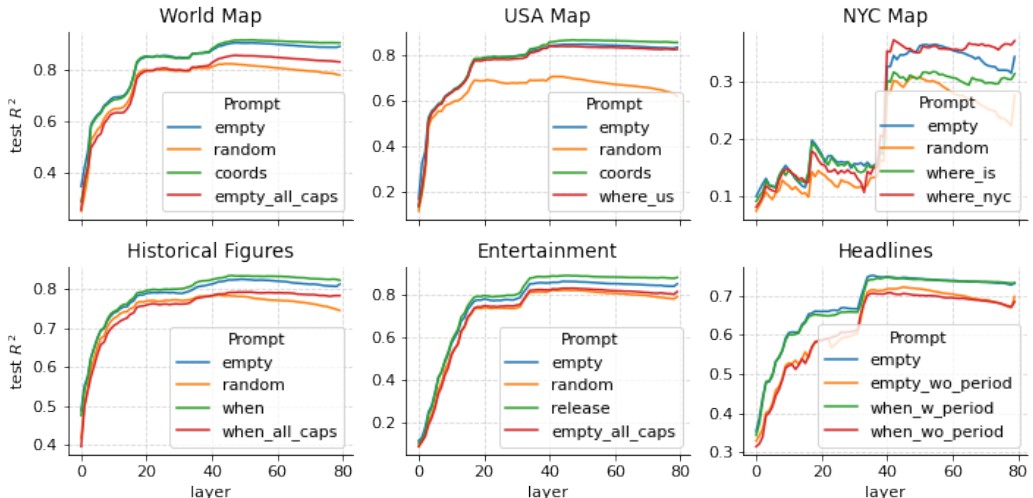

Figure 3: Out-of-sample $R^2$ when entity names are included in different prompts for Llama-2-70b.

To study this, we create new activation datasets where we prepend different prompts to each of the entity tokens, following a few basic themes. In all cases, we include an "empty" prompt containing nothing other than the entity tokens (and a beginning of sequence token). We then include a prompt which asks the model to recall the relevant fact, e.g., "What is the latitude and longitude of <place>" or "What was the release date of <author>'s <book>." For the United States and NYC datasets we also include versions of these prompts asking where in the US or NYC this location is, in an attempt to disambiguate common names of places (e.g. City Hall). As a baseline we include a prompt of 10 random tokens (sampled for each entity). To determine if we can obfuscate the subject, for some datasets we fully capitalize the names of all entities. Lastly, for the headlines dataset, we try probing on both the last token and on a period token appended to the headline.

We report results for the 70B model in Figure 3 and all models in Figure 8. We find that explicitly prompting the model for the information, or giving disambiguation hints like that a place is in the US or NYC, makes little to no difference in performance. However, we were surprised by the degree to which random distracting tokens degrades performance. Capitalizing the entities also degrades performance, though less severely and less surprisingly, as this likely interferes with "detokenizing" the entity (Elhage et al., 2022a; Gurnee et al., 2023; Geva et al., 2023). The one modification that did notably improve performance is probing on the period token following a headline, suggesting that periods are used to contain some summary information of the sentences they end.

Table 2: Out-of-sample $R^2$ of linear and nonlinear (one layer MLP) probes for all models and features at 60% layer depth.

| Model | Probe | World | USA | NYC | Historical | Entertainment | Headlines |
|---|---|---|---|---|---|---|---|
| | | | | | Dataset | | |
| Llama-2-7b | Linear | 0.881 | 0.799 | 0.219 | 0.785 | 0.788 | 0.564 |
| | MLP | 0.897 | 0.819 | 0.204 | 0.775 | 0.746 | 0.467 |
| Llama-2-13b | Linear | 0.896 | 0.825 | 0.237 | 0.804 | 0.806 | 0.645 |
| | MLP | 0.916 | 0.824 | 0.230 | 0.818 | 0.808 | 0.656 |
| Llama-2-70b | Linear | 0.911 | 0.864 | 0.359 | 0.835 | 0.885 | 0.746 |
| | MLP | 0.926 | 0.869 | 0.312 | 0.839 | 0.884 | 0.739 |

## 4  ROBUSTNESS CHECKS

The previous section has shown that the true point in time or space of diverse types of events or locations can be linearly recovered from the internal activations of the mid-to-late layers of LLMs. However, this does not imply if (or how) a model actually uses the feature direction learned by the probe, as the probe itself could be learning some linear combination of simpler features which are actually used by the model.

### 4.1  VERIFICATION VIA GENERALIZATION

**Block holdout generalization**   To illustrate a potential issue with our results, consider the task of representing the full world map. If the model has, as we expect it does, an almost orthogonal binary feature for is_in_country_X, then one could construct a high quality latitude (longitude) probe by summing these orthogonal feature vectors for each country with coefficient equal to the latitude (longitude) of that country. Assuming a place is in only one country, such a probe would place each entity at its country centroid. However, in this case, the model does not actually represent space, only country membership, and it is only the probe which learns the geometry of the different countries from the explicit supervision.

To better distinguish these cases, we analyze how the probes generalize when holding out specific blocks of data. In particular, we train a series of probes, where for each one, we hold out one country, state, borough, century, decade, or year for the world, USA, NYC, historical figure, entertainment, and headlines dataset respectively. We then evaluate the probes on the held out block of data. In Table 3, we report the average proximity error for the block of data when completely held out, compared to the error of the test points from that block in the default train-test split, averaged over all held out blocks.

We find that while generalization performance suffers, especially for the spatial datasets, it is clearly better than random. By plotting the predictions of the held out states or countries in Figures 11 and 12, a qualitatively clearer picture emerges. That is, the probe correctly generalizes by placing the points in the correct relative position (as measured by the angle between the true and predicted centroid) but not in their absolute position. We take this as weak evidence that the probes are extracting explicitly learned features by the model, but are memorizing the transformation from model coordinates to human coordinates. However, this does not fully rule out the underlying binary features hypothesis, as there could be a hierarchy of such features that do not follow country or decade boundaries.

Table 3: Average proximity error across blocks of data (e.g., countries, states, decades) when included in the training data compared to completely held out. Random performance is 0.5.

| Model | Block | Dataset | | | | | |
|-------|-------|-------|-----|-----|------------|---------------|-----------|
|       |       | World | USA | NYC | Historical | Entertainment | Headlines |
| Llama-2-7b | nominal | 0.071 | 0.144 | 0.331 | 0.129 | 0.147 | 0.258 |
|            | held out | 0.170 | 0.192 | 0.473 | 0.133 | 0.158 | 0.264 |
| Llama-2-13b | nominal | 0.068 | 0.144 | 0.319 | 0.121 | 0.141 | 0.223 |
|             | held out | 0.156 | 0.189 | 0.470 | 0.126 | 0.152 | 0.235 |
| Llama-2-70b | nominal | 0.071 | 0.121 | 0.262 | 0.115 | 0.105 | 0.182 |
|             | held out | 0.164 | 0.188 | 0.433 | 0.119 | 0.122 | 0.200 |

**Cross entity generalization**   Implicit in our discussion so far is the claim that the model represents the space or time coordinates of different types of entities (like cities or natural landmarks) in a unified manner. However, similar to the concern that a latitude probe could be a weighted sum of membership features, a latitude probe could also be the sum of different (orthogonal) directions for the latitudes of cities and for the latitudes of natural landmarks.

Similar to the above, we distinguish these hypotheses by training a series of probes where the train-test split is performed to hold out all points of a particular entity class.[1] Table 4 reports the proximity error for the entities in the default test split compared to when heldout, averaged over all such splits as before. The results suggest that the probes largely generalize across entity types, with the main exception of the entertainment dataset.[2]

Table 4: Average proximity error across entity subtypes (e.g. books and movies) when included in the training data compared to being fully held out. Random performance is 0.5.

| Model | Entity | Dataset | | | | | |
|---|---|---|---|---|---|---|---|
| | | World | USA | NYC | Historical | Entertainment | Headlines |
| Llama-2-7b | nominal | 0.120 | 0.206 | 0.313 | 0.164 | 0.224 | 0.199 |
| | held out | 0.151 | 0.262 | 0.367 | 0.168 | 0.305 | 0.289 |
| Llama-2-13b | nominal | 0.117 | 0.197 | 0.310 | 0.153 | 0.207 | 0.171 |
| | held out | 0.147 | 0.259 | 0.377 | 0.159 | 0.283 | 0.266 |
| Llama-2-70b | nominal | 0.113 | 0.173 | 0.266 | 0.149 | 0.159 | 0.144 |
| | held out | 0.147 | 0.203 | 0.322 | 0.149 | 0.271 | 0.219 |

## 4.2 DIMENSIONALITY REDUCTION

Despite being linear, our probes still have $d_{model}$ learnable parameters (ranging from 4096 to 8192 for the 7B to 70B models), enabling it to engage in substantial memorization. As a complementary form of evidence to the generalization experiments, we train probes with 2 to 3 orders of magnitude fewer parameters by projecting the activation datasets onto their $k$ largest principal components.

Figure 4 illustrates the test $R^2$ for probes trained on each model and dataset over a range of $k$ values, as compared to the performance of the full $d_{model}$-dimensional probe. We also report the test Spearman correlation in Figure 13 which increases much more rapidly with increasing $k$ than the $R^2$. Notably, the Spearman correlation only depends on the rank order of the predictions while $R^2$ also depends on their actual value. We view this gap as further evidence that the model explicitly represents space and time as these features must account for enough variance to be in the top dozen principal components, but that the probe requires more parameters to convert from the model's coordinate system to literal spatial coordinates or timestamps. We also observed that the first several principal components clustered the different entity types within the dataset, explaining why more than a few are needed.

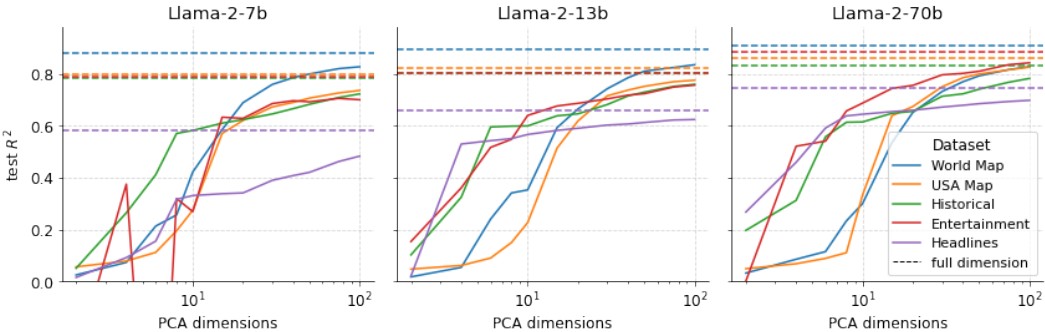

Figure 4: Test $R^2$ for probes trained on activations projected onto $k$ largest principal components for each dataset and model compared to training on the full activations.

---

[1]We only do this entities which do not make up the majority of the training data (e.g., as is the case with populated places for the world dataset and songs for the entertainment dataset) which is partially responsible for the discrepancies in the nominal cases for Tables 3 and 4.

[2]We note in this case the Spearman correlation is still high, suggesting this is an issue with bias generalization, as the different entity types are not uniformly distributed in time.

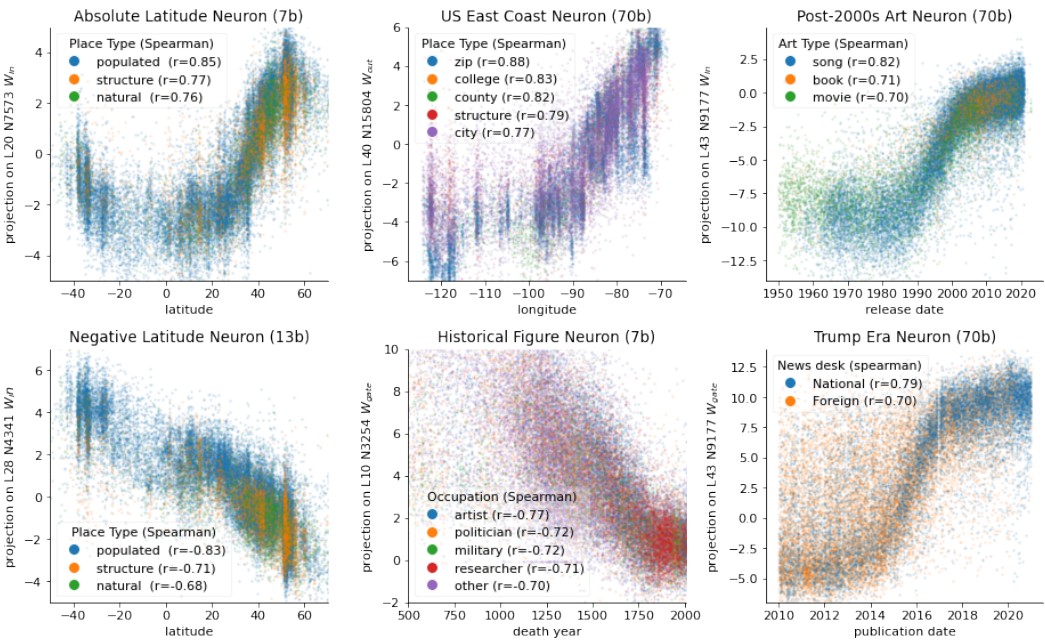

Figure 5: Space and time neurons in Llama-2 models. Depicts the result of projecting activation datasets onto neuron weights compared to true space or time coordinates with Spearman correlation by entity type.

## 5 SPACE AND TIME NEURONS

While the previous results are suggestive, none of our evidence directly shows that the model *uses* the features learned by the probe. To address this, we search for individual neurons with input or output weights that have high cosine similarity with the learned probe direction. That is, we search for neurons which read from or write to a direction similar to the one learned by the probe.

We find that when we project the activation datasets on to the weights of the most similar neurons, these neurons are indeed highly sensitive to the true location of entities in space or time (see Figure 5). In other words, there exist individual neurons within the model that are themselves fairly predictive feature probes. Moreover, these neurons are sensitive to all of the entity types within our datasets, providing stronger evidence for the claim these representations are unified.

If probes trained with explicit supervision are an approximate upper bound on the extent to which a model represents these spatial and temporal features, then the performance of individual neurons is a lower bound. In particular, we generally expect features to be distributed in superposition (Elhage et al., 2022b), making individual neurons the wrong level of analysis. Nevertheless, the existence of these individual neurons, which received no supervision other than from next-token prediction, is very strong evidence that the model has learned and makes use of spatial and temporal features.

We also perform a series of neuron ablation and intervention experiments in Appendix B to verify the importance of these neurons in spatial and temporal modeling.

## 6 RELATED WORK

**Linguistic Spatial Models** Prior work has shown that natural language encodes geographic information (Louwerse & Zwaan, 2009; Louwerse & Benesh, 2012) and that relative coordinates can be approximately recovered with simple techniques like multidimensional scaling, co-occurrence statistics, or probing word embeddings (Louwerse & Zwaan, 2009; Mikolov et al., 2013a; Gupta et al., 2015; Konkol et al., 2017). However, these studies only consider a few hundred well known cities and obtain fairly weak correlations. Most similar to our work is (Liétard et al., 2021) who probe word embeddings and small language models for the coordinates of global cities and whether

countries share a border, but conclude the amount of geographic information learned is "limited," likely because the largest model they study was 345M parameters (500x smaller than Llama 70B).

**Neural World Models** We consider a spatiotemporal model to be a necessary ingredient within a larger world model. The clearest evidence that such models are learnable from next-token prediction comes from GPT-style models trained on chess (Toshniwal et al., 2022) and Othello games (Li et al., 2022) which were shown to have explicit representations of the board and game state, with further work showing these representations are linear (Nanda et al., 2023). In true LLMs, Li et al. (2021) show that an entity's dynamic properties or relations can be linearly read out from representations at different points in the context. Abdou et al. (2021) and Patel & Pavlick (2021) show LLMs have representations that reflect perceptual and conceptual structure in color and spatial domains.

**Factual Recall** The point in time or space of an event or place is a particular kind of fact. Our investigation is informed by prior work on the mechanisms of factual recall in LLMs (Meng et al., 2022a;b; Geva et al., 2023) indicating that early-to-mid MLP layers are responsible for outputting information about factual subjects, typically on the last token of the subject. Many of these works also show linear structure, for example in the factuality of a statement (Burns et al., 2022) or in the structure of subject-object relations (Hernandez et al., 2023). To our knowledge, our work is unique in considering continuous facts.

**Interpretability** More broadly, our work draws upon many results and ideas from the interpretability literature (Räuker et al., 2023), especially in topics related to probing (Belinkov, 2022), BERTology (Rogers et al., 2021), the linearity hypothesis and superposition (Elhage et al., 2022b), and mechanistic interpretability (Olah et al., 2020). More specific results related to our work include Hanna et al. (2023) who find a circuit implementing greater-than in the context of years, and Goh et al. (2021) who find "region" neurons in multimodal models that resemble our space neurons.

## 7 DISCUSSION

We have demonstrated that LLMs learn linear representations of space and time that are unified across entity types and fairly robust to prompting, and that there exists individual neurons that are highly sensitive to these features. We conjecture, but do not show, these basic primitives underlie a more comprehensive causal world model used for inference and prediction.

Our analysis raises many interesting questions for future work. While we showed that it is possible to linearly reconstruct a sample's absolute position in space or time, and that some neurons use these probe directions, the true extent and structure of spatial and temporal representations remain unclear. We conjecture that the most canonical form of this structure is a discretized hierarchical mesh, where any sample is represented as a linear combination of its nearest basis points at each level of granularity. Moreover, the model can and does use this coordinate system to represent absolute position using the correct linear combination of basis directions in the same way a linear probe would. We expect that as models scale, this mesh is enhanced with more basis points, more scales of granularity (e.g. neighborhoods in cities), and more accurate mapping of entities to model coordinates (Michaud et al., 2023). This suggests future work on extracting representations in the model's coordinate system rather than trying to reconstruct human interpretable coordinates, perhaps with sparse autoencoders (Cunningham et al., 2023).

We also barely scratched the surface of understanding how these spatial and temporal models are learned, recalled, and used internally, or to what extent these representations exist within a more comprehensive world model. By looking across training checkpoints, it may be possible to localize a point in training when a model organizes constituent `is_in_place_X` features into a coherent geometry or else conclude this process is gradual (Liu et al., 2021). We expect that the model components which construct these representations are similar or identical to those for factual recall (Meng et al., 2022a; Geva et al., 2023).

Finally, we note that the representation of space and time has received much more attention in biological neural networks than artificial ones (Buzsáki & Llinás, 2017; Schonhaut et al., 2023). Place and grid cells (O'Keefe & Dostrovsky, 1971; Hafting et al., 2005) in particular are among the most well-studied in the brain and may be a fruitful source of inspiration for future work on LLMs.

ACKNOWLEDGEMENTS

The authors would like to thank Sam Marks, Eric Michaud, Ziming Liu, Janice Yang, and especially Neel Nanda for their helpful discussions and feedback. W.G. was supported by Dimitris Bertsimas and an Open Philanthropy early career grant through the course of this work.

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

## A    DATASETS

We describe the construction and post-processing of our data in more detail in addition to known limitations. All datasets and code are available at `https://github.com/wesg52/world-models`.

**World Places**    We ran three separate queries to obtain the names, location, country, and associated Wikipedia article of all physical places, natural places, and structures within the DBPedia database Lehmann et al. (2015). Using the Wikipedia article link, we joined this information with data from the Wikipedia pageview statistics database [3] to query how many times this page was accessed over 2018-2020. We use this as a proxy for whether we should expect an LLM to know of this place or not, and filter those with less than 5000 views over this time period.

Several limitation are worth highlighting. First, our data only comes from English Wikipedia, and hence is skewed towards the Anglosphere. Additionally, the distribution of entity types is not uniform, e.g. we noticed the United Kingdom has many more railway stations than any other country, which could introduce unwanted correlations in the data that may affect the probes. Finally, about 25% of the samples had some sort of state or province modifier at the end like "Dallas County, Iowa". Because many of these locations were more obscure or would be ambiguous without the modifier, so we chose to rearrange the string to be of the from "Iowa's Dallas County" such the entity is disambiguated but that we are not probing on a token that is a common country or state name.

**USA Places**    The United States places dataset uses structures and natural places within the US from the world places dataset as a starting point, in addition to another DBPedia for US colleges. We then collect the name, population total, and state for every county[4], zipcode [5], and city [6] from a census data aggregator. We then remove all duplicate county or city names (there are 31 Washington counties in the US!), though we keep any duplicates that have 2x the population has the next largest place of the same name. We also filter out cities with fewer than 500 people, zipcodes with fewer than 10000 (or with population density greater than 50 and population greater than 2000), and any place not in the lower 48 contiguous states (or Washington D.C.).

**NYC Places**    Our New York City dataset is adapted from the NYC Open Date points of interest dataset (NYC OpenData, 2023) containing the names of locations tracked by the city government. This includes the names of schools, places of worship, transit locations, important roads or bridges, government buildings, public housing, and more. Each of these places comes with a complex ID for locations comprised of multiple such buildings (e.g. New York University or LaGuardia airport). We construct our test train splits to make sure that all locations within the same complex are put in the same split to avoid test-train leakage. We filtered out a large number of locations describing the position of bouys in the multiple waterways surrounding NYC.

**Historical Figures**    Our historical figures dataset contains the names and occupation of historical figures who died between 1000BC-2000AD adapted from (Annamoradnejad & Annamoradnejad, 2022). We filtered the dataset to only contain the 350 most famous people who died from each decade, imperfectly measured by the index of their Wikidata entity identifier.

---

[3] `https://en.wikipedia.org/wiki/Wikipedia:Pageview_statistics`
[4] `https://simplemaps.com/data/us-counties`
[5] `https://simplemaps.com/data/us-zips`
[6] `https://simplemaps.com/data/us-cities`

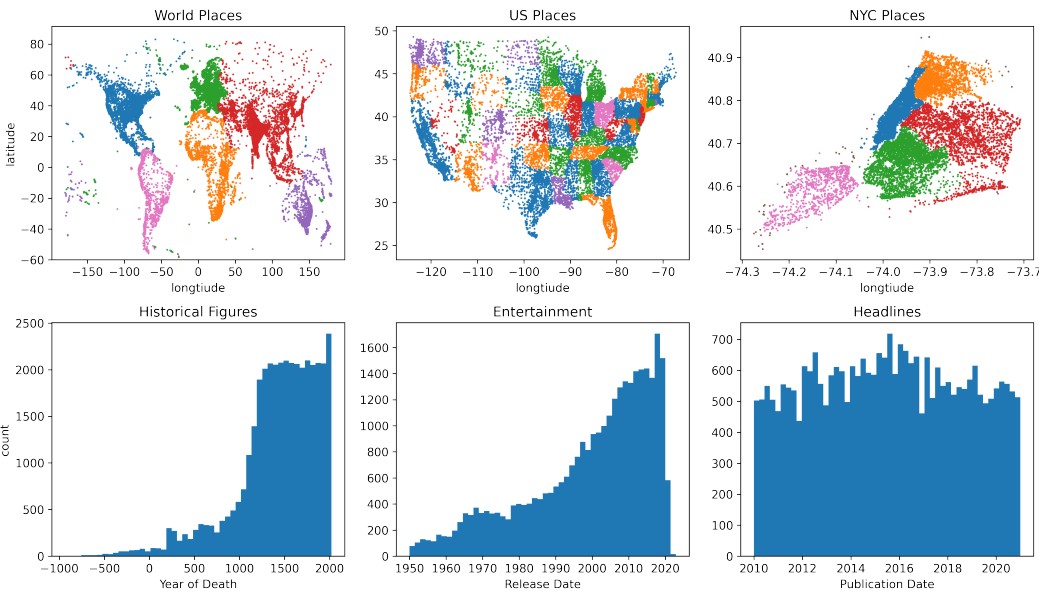

Figure 6: Distribution of samples in space or time for all datasets.

**Art and Entertainment**   Our art and entertainment dataset consists of the names of songs, movies, and books with their corresponding artist, director, and author release date. We constructed this dataset from DBpedia and similarly filtered out entities which had received less than 5000 page views over 2018-2020. Because many songs or books have fairly generic titles, we include the creator's name in the prompt to disambiguate (e.g. "Stephen Kings' It" for the empty prompt). However, because some artists or authors release many songs or books, we sample our test-train split by creator to avoid leakage.

**Headlines**   Our headlines dataset is adapted from a scrape of all New York Times headlines of the past 30 years (Bandy, 2021). In an attempt to filter out headlines which do not describe an event that could be localized in time, we employ a number of strategies. First we filter anything which is not within the first 10 pages of the print edition. Second we filter out articles that don't come from the Foreign, National, Politics, Washington, or Obits news desks. Third we removed any titles that contained a question mark.

## B  NEURON ABLATIONS AND INTERVENTIONS

To better understand the role of space and time neurons in LLMs, we conduct several neuron ablation and intervention experiments.

**Time Intervention**  We study the effect of intervening on a single time neuron (L19.3610; correlation with art and entertainment release date of 0.77) within Llama-2-7b. Given a prompt of the form `<media> by <creator> was written in 19`, we pin the activation of the time neuron on all tokens and sweep over a range of pinned values, and track the predicted probability of the top five tokens. Results are depicted in Figure 7 and show that just adjusting the time neuron activation can change the next token prediction in all cases.

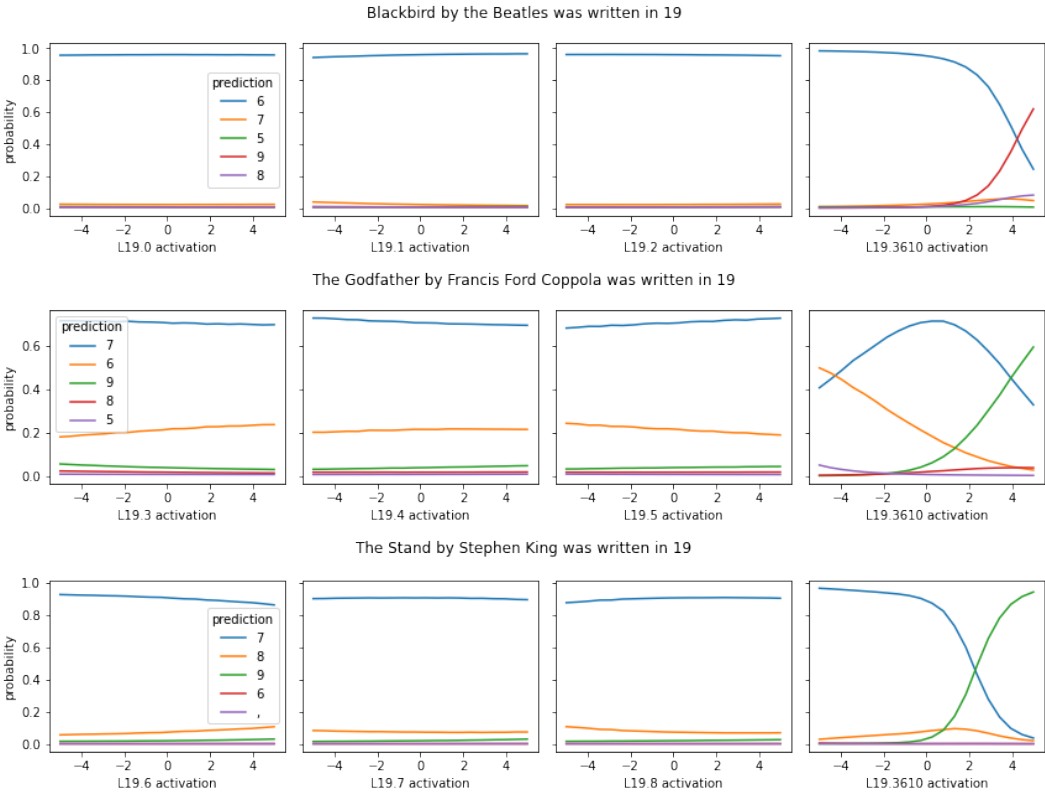

Figure 7: Prediction of decade of publication for a famous song, movie, and book when a time neuron (L19.3610) is pinned to a particular value, compared to 9 random neurons within the same layer (L19.[0-8]) of Llama-2-7b.

**Neuron Ablations**  We also study the effect of zero ablating neurons, and the contexts for which the loss increases the most. For a subset of Wikipedia which includes articles corresponding to world places and contemporary art and entertainment, we first run Llama-2-7b as normal and record the loss. Then, for two space neurons and two time neurons, we run the model with the neuron activation pinned to 0 (we always pin exactly one neuron to 0). For each neuron, we report the top 10 contexts in which the loss most increased for the next token prediction in Tables 5-8.

| context | true token | loss increase |
|---|---|---|
| Bom Jesus has a rather dry tropical savanna climate (Köppen | Aw | 2.107 |
| line tropical monsoon/humid subtropical climate (Köppen | Am | 2.035 |
| of . The highest temperatures are reached at the end of the dry season in | March | 1.960 |
| 8.9 °C. In January, the average temperature is 1 | 8 | 1.930 |
| a Tropical wet-and-dry climate (Köppen climate classification | Aw | 1.876 |
| Goroka has a relatively cool tropical monsoon climate (Köppen | Am | 1.854 |
| ie range from 26.4 °F in January to 7 | 0 | 1.835 |
| wet summers and warm, very wet winters (Köppen climate classification | Am | 1.807 |
| tropical wet and dry climate/semi-arid climate (Köppen | Aw | 1.783 |
| ably mild, tropical-maritime climate, (Köppen climate classification | Aw | 1.762 |

Table 5: Contexts with the highest loss when ablating space neuron L20.7573 from Llama-2-7b.

| context | true token | loss increase |
|---|---|---|
| mark is Mount Etna, one of the tallest active volcanoes in | Europe | 1.971 |
| 2,800 years, making it one of the oldest cities in | Europe | 1.676 |
| keeper with 63 caps for Portugal including participation in the 198 | 4 | 1.631 |
| 15. Tenerife also has the largest number of endemic species in | Europe | 1.512 |
| Georgia to the south-west. Mount Elbrus, the highest mountain in | Europe | 1.246 |
| At one point, the village boasted the longest aluminium rolling mill in | Western | 1.219 |
| centre and the leading economic hub of the Iberian Peninsula and of | Southern | 1.181 |
| atican City, a sovereign state—and possibly the second largest in | Europe | 1.103 |
| name. This is because the British Isles were likely repopulated from the | I | 1.082 |
| Category 4 stadium by UEFA, hosted matches at the 199 | 8 | 1.072 |

Table 6: Contexts with the highest loss when ablating space neuron L20.7423 from Llama-2-7b.

| context | true token | loss increase |
|---|---|---|
| was released in June 1993 as the fourth single from the album | He | 2.254 |
| was released in November 1992 as the second single from her album | He | 1.973 |
| 93. Yearwood's version was the third single from her album | He | 1.749 |
| Hot Country Singles & Tracks chart, behind Shania Twain's ¨ | Any | 1.574 |
| was released in February 1992 as the third single from the album | What | 1.559 |
| and provided additional production on her singles "Like A Prayer" and " | Express | 1.481 |
| was released in May 1992 as the fourth single from the album | What | 1.367 |
| filmmaker Ramesh Aravind in Telugu cinema. P. | L | 1.328 |
| 993 by record label Columbia as the second single from their second studio album | Gold | 1.272 |
| Gessle for the duo's 1991 album, | Joy | 1.253 |

Table 7: Contexts with the highest loss when ablating time neuron L18.9387 from Llama-2-7b.

| context | true token | loss increase |
|---|---|---|
| 016 as the third single from Reyes debut studio album, Louder | ! | 1.082 |
| as the second radio single in support of the band's third studio album, | Life | 0.996 |
| Song, but ultimately lost the award to Barbra Streisand's " | The | 0.965 |
| Released as the first single from the group's seventh studio album, | Super | 0.961 |
| original 30 Carry On films (1958–19 | 7 | 0.912 |
| . After listening to No Doubt's 2002 single " | Under | 0.867 |
| ix9ine for his debut mixtape Day69 (201 | 8 | 0.866 |
| BA. It was originally featured on the group's fifth studio album, The | Album | 0.864 |
| ck that appeared in the Porky Pig cartoons It's | an | 0.805 |
| was written by Andrew Lloyd Webber and Tim Rice and produced by | Fel | 0.805 |

Table 8: Contexts with the highest loss when ablating time neuron L19.3610 from Llama-2-7b.

# C    ADDITIONAL RESULTS

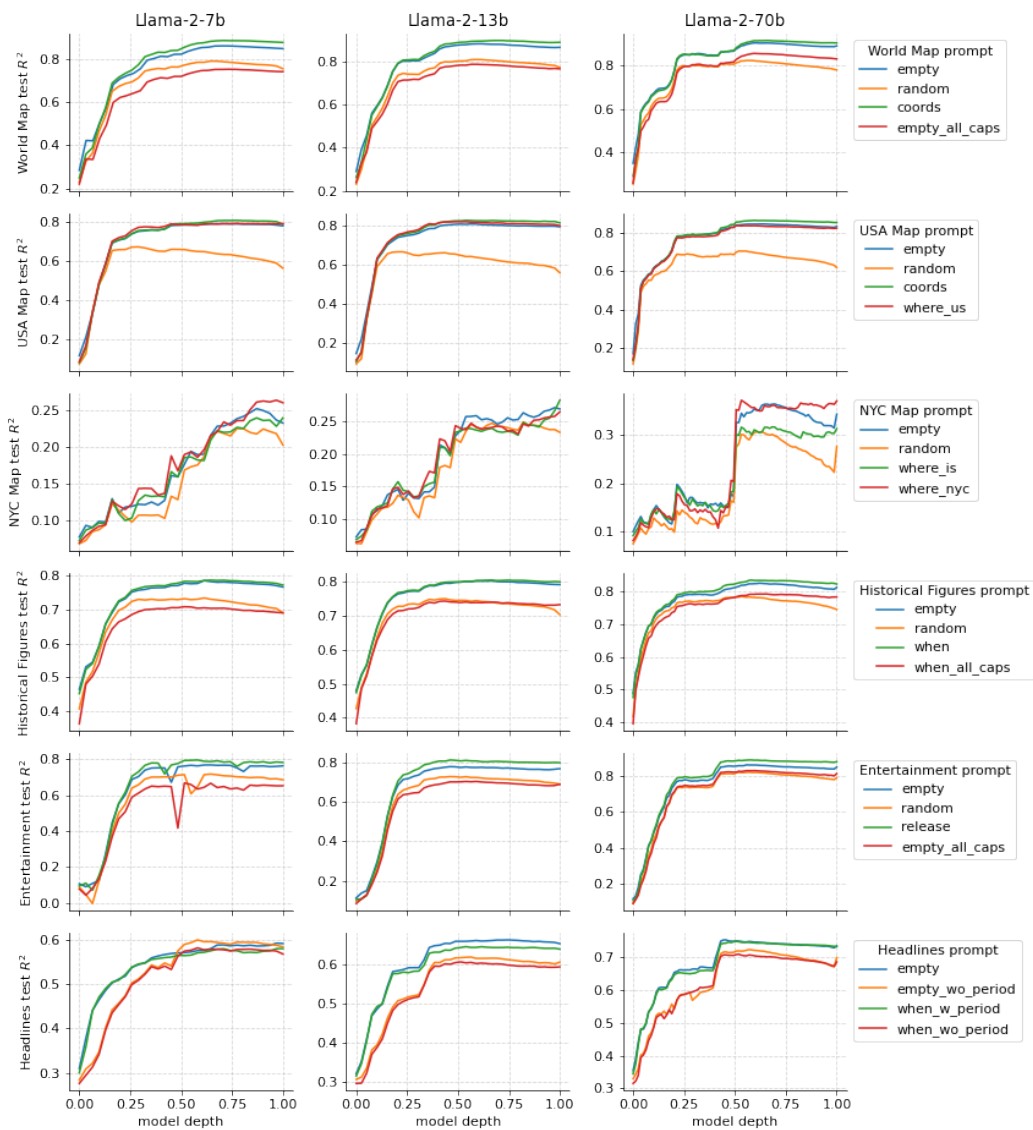

Figure 8: Out-of-sample $R^2$ when entity names are included in different prompts for all models.

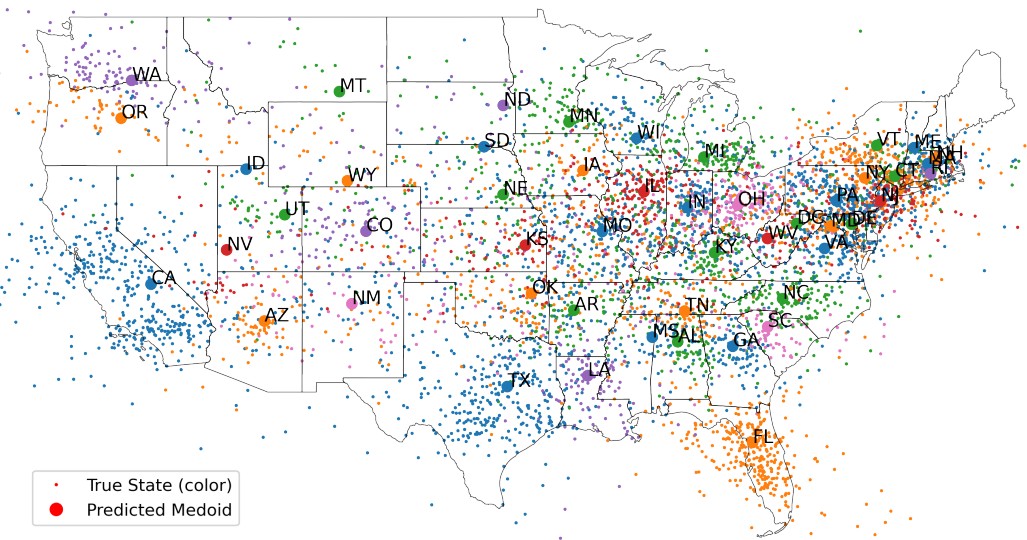

Figure 9: Llama-2-70b layer 50 model of the United states. Points are projections of activations of heldout US places onto learned latitude and longitude directions colored by true state, with median state prediction enlarged. All points depicted are from the test set.

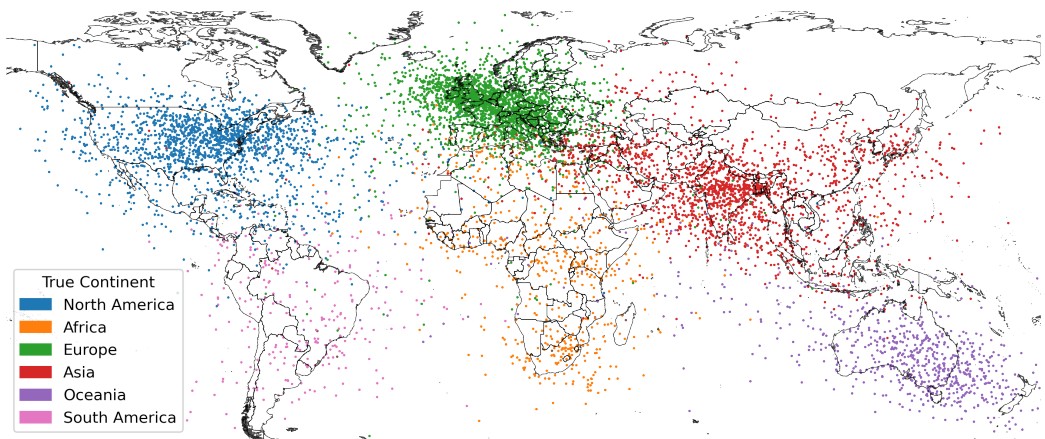

Figure 10: Llama-2-70b layer 50 model of the world. Points are projections of activations of heldout world places onto learned latitude and longitude directions colored by true continent. All points depicted are from the test set.

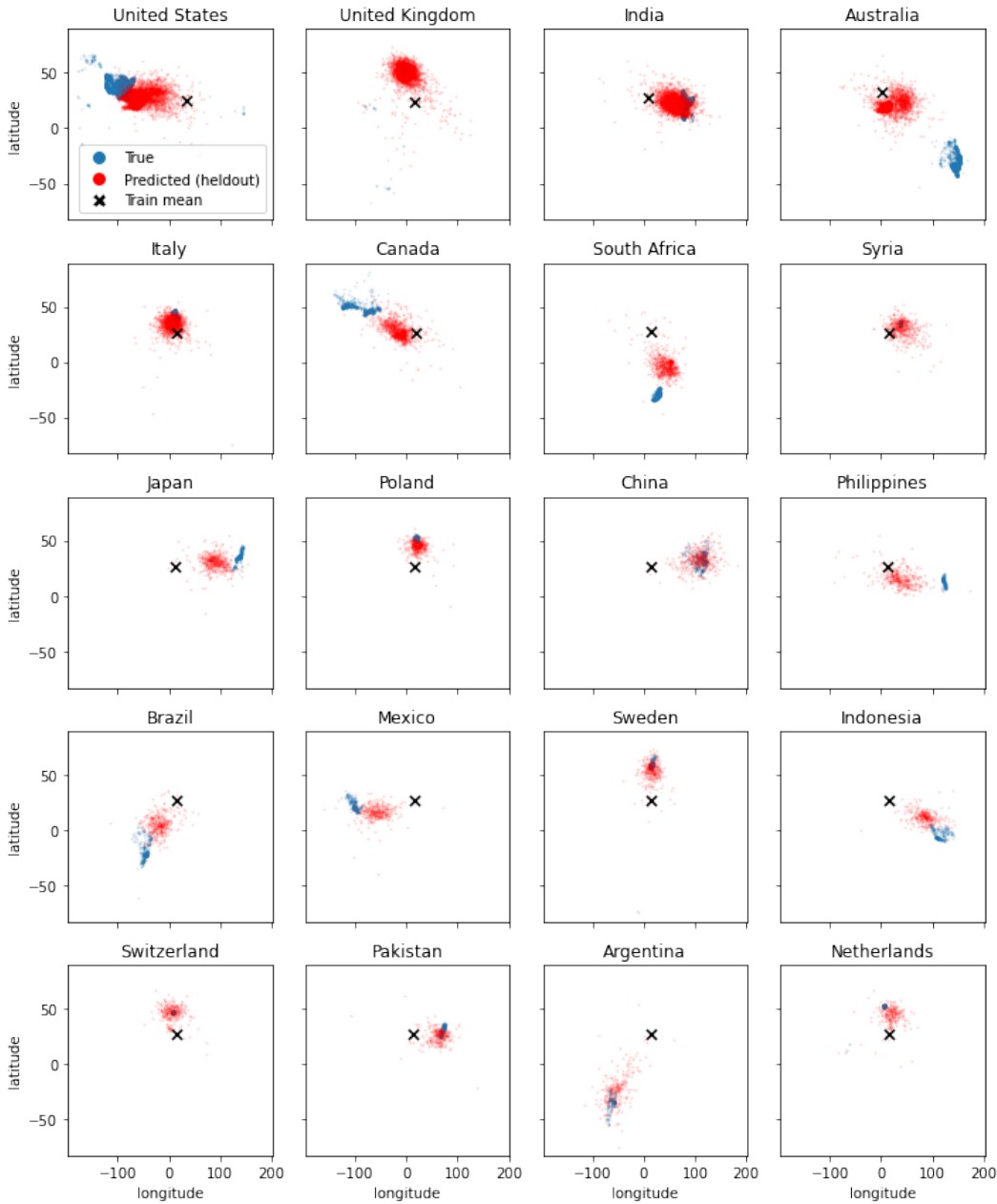

Figure 11: Out-of-sample predictions for each country when the probe training data contains no samples from the country as compared to true locations and the mean of the training data. The results imply that the learned feature direction correctly generalizes to the relative position of a country but that the probes memorizes the absolute positions.

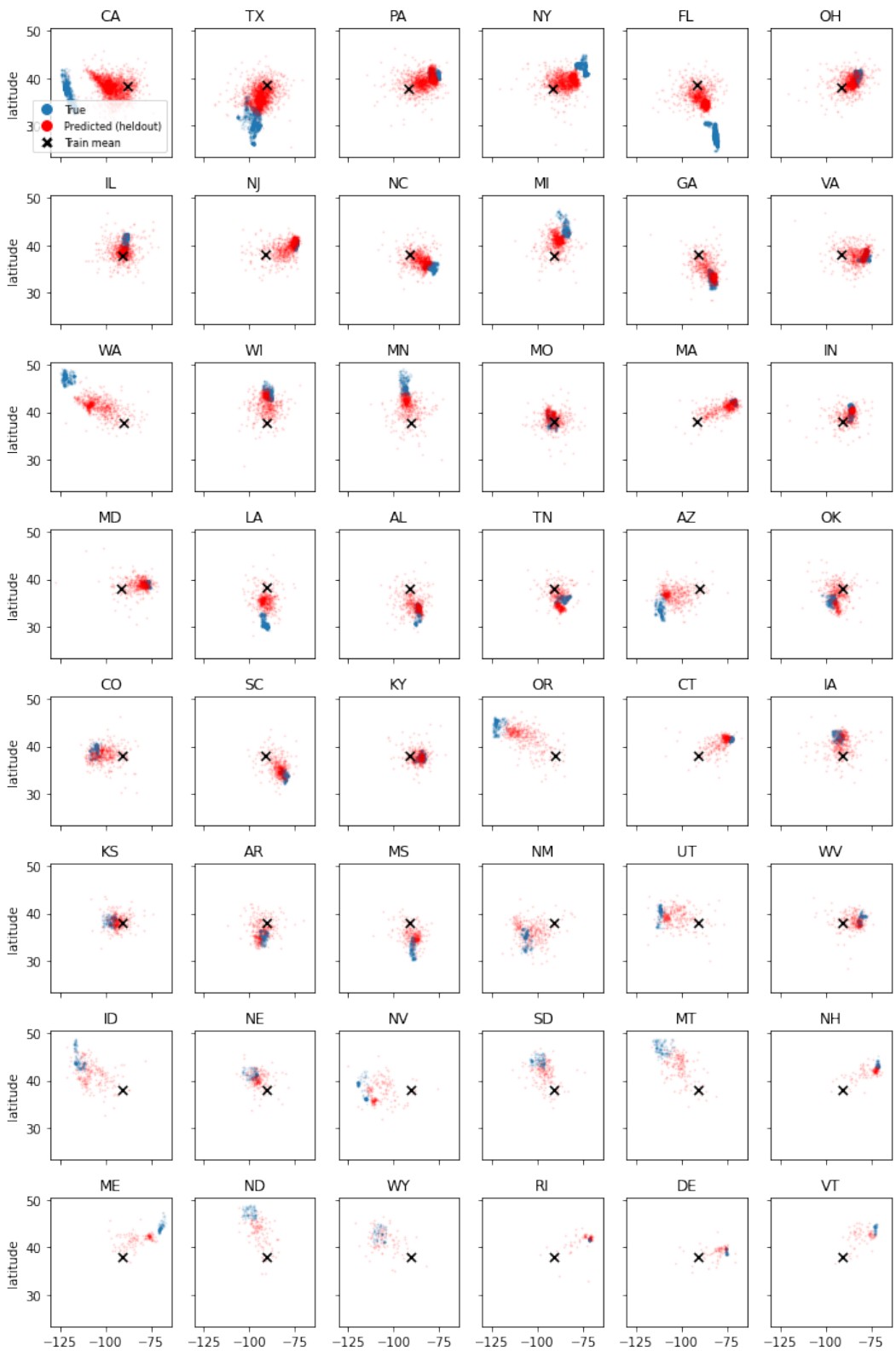

Figure 12: Out-of-sample predictions for each state when the probe training data contains no samples from the state as compared to true locations and the mean of the training data. The results imply that the learned feature direction correctly generalizes to the relative position of a country but that the probes memorizes the absolute positions.

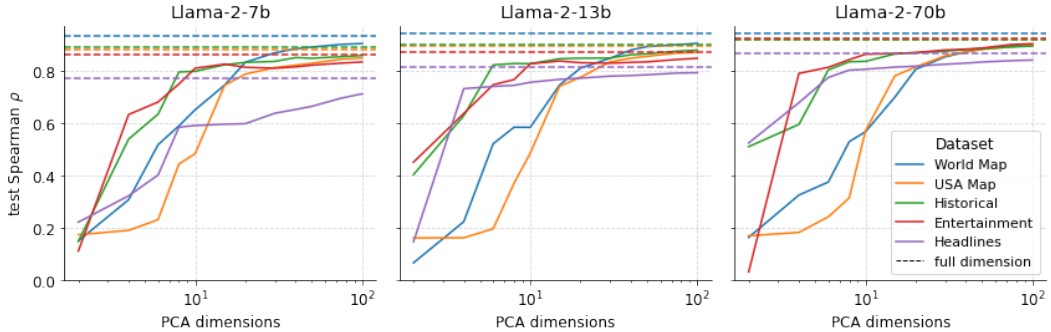

Figure 13: Test Spearman rank correlation for probes trained on activations projected onto $k$ largest principal components.

