# OpenReview forum: "Language Models Represent Space and Time"
_ICLR.cc/2024/Conference — ICLR 2024 poster_

### Official Review · Reviewer_MD6k · 2023-11-01

**Soundness:** 3 good
**Presentation:** 4 excellent
**Contribution:** 3 good
**Rating:** 8
**Confidence:** 4

**Summary:**

This paper probes language models (specifically the Llama-2 type) for representations of spatial location of place names at different levels of granularity, as well as placement in time of people, products and events.
Activations in language models are mapped to explicit geographical and temporal coordinates via linear probes. The paper finds that the tested language models in general have robust representations of both spatial and temporal location; locations within New York City however are mapped with relatively lower accuracy. Additionally the paper localizes specific neurons especially sensitive to location in either space or time.

**Strengths:**

The works uses straightforward methods and well thought out experimental setup. Experiments are exhaustive. The results are presented in a clear and easy to follow fashion.

**Weaknesses:**

The main problem with this work is the serious lack of awareness and engagement with very similar work carried out not so long ago [1,2]. As a results, the paper doesn't acknowledge that qualitatively similar results obtain even in extremely simplistic models applied to text such as SVD and LSA, and overinterprets the results as evidence of the language models analyzed possessing "a coherent model of the data generating process—a world model".
If the spatial and temporal representations found here count as a world  model, it's a a very partial and almost trivial one. It certainly is very far from a complete model of the process generating textual data in general. The paper would be much improved if it seriously toned down these overly dramatic claims.

A minor methodological quibble: longitude wraps around, so linear correlation is not an ideal measure here.

[1] Louwerse, M.M., & Zwaan, R.A. (2009). Language Encodes Geographical Information. Cognitive science, 33 1, 51-73 .

[2] Louwerse, M.M., & Benesh, N. (2012). Representing Spatial Structure Through Maps and Language: Lord of the Rings Encodes the Spatial Structure of Middle Earth. Cognitive science, 36 8, 1556-69 .

**Questions:**

Proximity error is defined as "fraction of predictions closer to the target point than the actual prediction". I don't understand what this means; should this read "fraction of datapoints closer to the target point than the actual prediction"?

I don't understand how the mere presence of specific neurons correlated to spatial/temporal probes counts as evidence that these representations are used by the model. A more convincing evidence would involve some intervention on these neurons.

---

> ### Author Response · Authors · 2023-11-20
> **Response to MD6k 1/2**
>
> Thank you for the comments.
>
> > The main problem with this work is the serious lack of awareness and engagement with very similar work carried out not so long ago [1,2]
>
> Thank you for bringing these references to our attention. We reorganized the related work section and added a new paragraph on “Linguistic Spatial Models” and included these very relevant and interesting works (among others pointed out by other reviewers).
>
> While we think these prior works are great proofs of concept that geographic structure is in principle learnable from language, we respectfully disagree with the characterization that they are “very similar,” beyond the topic. Eg, [1] used a dataset of just 50 well known US cities and [2] used 32 fictional cities, with both obtaining fairly weak R^2 values (around 0.5).
>
> We think our work is differentiated from all prior work noted in this new section by:
> * Also considering temporal representations
> * Studying models with more than XXXM parameters (we go up to 70B which we show is quite important) and using far more probing data (at least 10x more)
> * Studying spatial and temporal representations at different spatiotemporal scales (eg New York City, US based, and world map)
> * Studying how different types of entities are represented (eg, natural landmarks and physical structures), whereas all prior work mostly focuses on cities.
> * Studying aspects unique to language models, like where the representations form in the model, or how sensitive the representations are to prompting.
> * Beginning to dive into model internals, and identifying individual model components which respond to spatial and temporal representations.
>
> Another important contribution of our work is the datasets we created, each of which on its own would be 10x more data than any prior study used, and we created 6 such datasets spanning diverse entity types and scales. We make these datasets publicly available in our github repository and believe they will enable important future work on understanding spatial and temporal reasoning and interpretability.
>
> >  overinterprets the results as evidence of the language models analyzed possessing "a coherent model of the data generating process—a world model" If the spatial and temporal representations found here count as a world model, it's a a very partial and almost trivial one. It certainly is very far from a complete model of the process generating textual data in general.
>
> We agree that what we show is very far from a complete model of the data generating process and that this is a very incomplete world model and did not mean to imply that it was. However, we think that coherent spatial and temporal representations are necessary for a more complete world model, and therefore find our results suggestive that modern LLMs might contain such an object, and more research in this direction is worthwhile. We have tried to make these claims more clear (see below).
>
> > The paper would be much improved if it seriously toned down these overly dramatic claims.
>
> We apologize for being overzealous in our initial wording of our claims. In the revision we attempted to be more clear about what we actually show, and what we merely conjecture, and in general add more qualification. The largest changes:
>
> * Last sentence of abstract “Our analysis demonstrates that modern LLMs acquire structured knowledge about fundamental dimensions such as space and time, supporting the view that they learn not merely superficial statistics, but literal world models.” -> “While further investigation is needed, our results suggest modern LLMs learn rich spatiotemporal representations of the real world and possess basic ingredients of a world model.”
> * In the intro we add: “While such spatiotemporal representations do not constitute a dynamic causal world model in their own right, having coherent multi-scale representations of space and time are basic primitives required of a more comprehensive model.”
> * In the discussion we add “We conjecture, but do not show, these basic primitives underlie a more comprehensive causal world model used for inference and prediction.”

---

> ### Author Response · Authors · 2023-11-20
> **Response to MD6k 2/2**
>
> > A minor methodological quibble: longitude wraps around, so linear correlation is not an ideal measure here.
>
> Indeed, this is not ideal. Ultimately we thought it would be most compelling to find these representations using the simplest possible method but for longitude in particular this is less principled. However, in practice it mostly does not matter given the dearth of points in the pacific ocean.
>
> > Proximity error is defined as "fraction of predictions closer to the target point than the actual prediction". I don't understand what this means; should this read "fraction of datapoints closer to the target point than the actual prediction"?
>
> It is the fraction of predicted datapoints. In other words, what fraction of entities were predicted to be closer to the target point than the prediction of the target entity. We reworded the text to clarify – thank you for raising the issue.
>
> > I don't understand how the mere presence of specific neurons correlated to spatial/temporal probes counts as evidence that these representations are used by the model.
>
> In addition to the neuron activations being quite correlated to the actual space/time coordinates of entities, the neuron input weights have high cosine similarity to the probe weights. This demonstrates the neurons are “reading” from a direction in the residual stream similar to the probe direction and hence is sensitive to some of the same information the probe is using at a level which is far above what one would observe from random chance.
>
> > A more convincing evidence would involve some intervention on these neurons.
>
> We performed several ablation studies which we describe in the comment to all reviewers and added to the appendix of the revised article.

---

### Official Review · Reviewer_XR65 · 2023-11-02

**Soundness:** 3 good
**Presentation:** 4 excellent
**Contribution:** 2 fair
**Rating:** 5
**Confidence:** 4

**Summary:**

This paper studies the spatial and temporal data representations for LLMs. It utilizes three spatial datasets (world, US, NYC places) and three temporal datasets (historical figures, artworks, news headlines) to analyze the internal representations in the Llama-2 model family. The study finds that LLMs develop linear, robust representations of space and time, unified across various entity types. It also identifies specific "space neurons" and "time neurons" within these models, indicating that LLMs form structured knowledge about space and time. The research, which employs linear regression probes on model activations to predict real-world locations or times, underscores the potential of LLMs in developing complex world models, impacting their robustness and application in AI systems.

**Strengths:**

1. The paper presents an exploration into the internal workings of large language models (LLMs), particularly focusing on how these models internalize and represent continuous variables including spatial and temporal dimensions.
2. The investigation is well-conducted, employing rigorous experiments and analysis methods to support that the probing results are not merely superficial statistics. The identification of specific "space neurons" and "time neurons" within these models is an innovative aspect to understand LLMs.
3. This study provides evidence that large language models possess an intrinsic comprehension of world models. This insight can enhance researchers' understanding of how information is represented within large language models.

**Weaknesses:**

1. One of the paper's limitations is its potential overlap with findings already established in the word embedding literature. Earlier works in this area, such as those by Mikolov et al. (2013), have demonstrated that simpler word embedding models can capture relational information and regularities in a linear fashion. This raises the question of whether the findings in this paper are genuinely novel or simply an extension of what is already known about linear representations in language models. The paper could strengthen its contribution by more directly addressing how its findings with large language models (LLMs) differ significantly from those with simpler models, and by delving deeper into the implications of these differences for the field.
2. The assertion that LLMs learn literal 'world models' may be overstated. The concept of a 'world model' in the context of AI and cognitive science typically refers to a comprehensive and dynamic representation of the external world, including an understanding of causal relationships and the ability to predict future states. The paper's findings, while impressive, primarily demonstrate that LLMs can encode spatial and temporal information linearly. This is a far cry from the richer and more dynamic conception of a world model. A more accurate claim might be that LLMs are capable of forming structured and useful representations of spatial and temporal information, but these do not necessarily constitute comprehensive world models. Further exploration into how these representations are utilized by LLMs in real-world tasks, and how they compare to human cognitive processes, could provide a more nuanced understanding of their nature and limitations.
3. Though the analysis experiments are useful and abundant within the given datasets and LLMs, the scope of data and experiments is limited. In particular, the datasets used, while diverse, primarily represent well-known entities and locations. This could limit the generalizability of the findings to less common or more ambiguous entities. The study largely focuses on data in English and from a Western perspective. Exploring how these models represent space and time in other languages and cultural contexts could provide valuable insights into their versatility and limitations. Also, the study is conducted on the Llama-2 model family. Testing the findings across different models and architectures would strengthen the argument that these capabilities are inherent to LLMs in general, rather than specific to a particular model.

**Questions:**

- What if the space or time neurons are removed/masked? How would it influence the models’ understanding of space or time?

---

> ### Author Response · Authors · 2023-11-20
> **XR65 Response 1/2**
>
> Thank you for your thoughtful comments!
>
> > W1. One of the paper's limitations is its potential overlap with findings already established in the word embedding literature.
>
> We agree this paper is inspired by prior work on embeddings, and that the observation that there exists linear structure in learned representations that reflect real world structure is an old and well known fact.
>
> However, static word embeddings are much less interesting objects of study and these past works mostly show that there exists structure in the most simple setup. For example, Mikolov et al. (2013) took 11 countries and their capitals in the northwestern hemispheres and did a PCA which showed a roughly correct ordering with no quantitative metrics. We feel that our expansion to almost one hundred thousand places on three scales of space and time and more thorough quantitative analysis provides a major step forward compared to this interesting prior work, and that this is of importance given recent discourse about the extent to which LLMs are "stochastic parrots".
>
> > W1. The paper could strengthen its contribution by more directly addressing how its findings with large language models (LLMs) differ significantly from those with simpler models, and by delving deeper into the implications of these differences for the field.
>
> Specifically, we think our work is novel compared to work on similar topics by:
> * Also considering temporal representations
> * Studying models with more than XXXM parameters (we go up to 70B which we show is quite important) and using far more probing data (at least 10x more)
> * Studying spatial and temporal representations at different spatiotemporal scales (eg New York City, US based, and world map)
> * Studying how different types of entities are represented (eg, natural landmarks and physical structures), whereas all prior work mostly focuses on cities.
> * Studying aspects unique to language models, like where the representations form in the model, or how sensitive the representations are to prompting.
> * Beginning to dive into model internals, and identifying individual model components which respond to spatial and temporal representations.
>
> Furthermore, work on embeddings is typically restricted only to places which are tokenized as a single word, and hence the datasets are far smaller (and likely even less globally diverse). Another important contribution of our work is the datasets we created, each of which on its own would be 10x more data than any prior study used, and we created 6 such datasets spanning diverse entity types and scales. We make these datasets publicly available in our github repository and believe they will enable important future work on understanding spatial and temporal reasoning and interpretability.
>
> > W2. The assertion that LLMs learn literal 'world models' may be overstated.
>
> We meant “literal world models” to mean “a literal model of the world” which, in hindsight, we agree was too glib - we wish to apologize for this overstatement. That said, we still believe such representations are a necessary condition for a more stringent conception of a world model, and we think our results are an important and foundational step in understanding to what degree there exists such a model. We edited the abstract, introduction, and conclusion in the submitted revision to make our claims more specific, in line with what you recommend.
>
> > W2. Further exploration into how these representations are utilized by LLMs in real-world tasks...
>
> To better understand how these representations are used, and to answer Q1, we perform neuron ablation experiments. See the change log comment and appendix B for analysis.
>
> > W2. and how they compare to human cognitive processes
>
> We think experimental comparisons to how humans represent space and time is out of scope for this study, but we do think this is an exciting area for future work (as discussed in the last paragraph of the discussion section) and note that just recently “A neural code for time and space in the human brain” by Schonhaut et al. was published in Cell two weeks ago and show the existence of time and place cells similar to our space and time neurons! We added this reference to the last paragraph in the discussion.

---

> ### Author Response · Authors · 2023-11-20
> **XR65 Response 2/2**
>
> > W3. In particular, the datasets used, while diverse, primarily represent well-known entities and locations.
>
> We respectfully disagree that the datasets present primarily well-known entities and locations, though we thank the reviewer for pointing out our lack of clarity on this point. While we expect the contents of Table 1 to be familiar to most readers, most of the datasets contain fairly obscure entities that we would expect the average reader to only know of a very small fraction.
> For example, here are the 10 world places with the median number of page views (ie, half of our world places dataset has a number of wikipedia page views less than these):
> 'San Isidro, Parañaque', 'Linslade', 'Khandoli Dam', 'Fucecchio', 'Rimatara', 'Bardon Mill', 'Poulaphouca', 'Paulsgrove', 'Mutton Island, County Galway', 'Gargnano'
>
> > The study largely focuses on data in English and from a Western perspective.
>
> We agree that our study is over representative of English data from a Western perspective, but this a reflection of the availability of web data, and the training data of the Llama models (english only). Hence, we agree this is a limitation, but not one that can be easily addressed (and is discussed in the appendix on datasets).
>
> > Also, the study is conducted on the Llama-2 model family. Testing the findings across different models and architectures would strengthen the argument that these capabilities are inherent to LLMs in general, rather than specific to a particular model.
>
> To validate the generality of our findings as suggested, we reran our main analysis a totally different suite of models:the Pythia model suite. The results replicate, where the most notable observation is the fairly clean scaling of probe performance vs. model size. We believe this indicates that LLMs do represent space and time, but that scale is especially important (another key aspect missing from past studies).
>
> > What if the space or time neurons are removed/masked? How would it influence the models’ understanding of space or time?
>
> We performed several ablation studies which we describe in the comment to all reviewers and added to the appendix of the revised article.

---

### Official Review · Reviewer_eYoU · 2023-11-04

**Soundness:** 3 good
**Presentation:** 4 excellent
**Contribution:** 3 good
**Rating:** 6
**Confidence:** 4

**Summary:**

This work examines whether representations of entities from the Llama-2 family language models represent important properties such as spatial coordinates for famous landmarks and years for famous people and events. This is done via training linear ridge regression probes on top of the representations extracted from activations at individual layers of the models. The results are that across several datasets, the LLM representations encode space and time information, and that these representations improve through the first half of the model layers and are then stable through the remainder of the model.

**Strengths:**

- examine an interesting and timely question that will be relevant to the ICLR community
- well-designed experiments to rule out confounding factors
- investigate several different open source models

**Weaknesses:**

W1. This work uses established methods for probing and there is no methodological innovation, though this is on its own not a deal breaker

W2. The research question is motivated as trying to study whether LLMs build a world model, but it's not clear to me why learning important properties of famous landmarks, such as location, and of famous events, such as years, are a sign of a "world model". My guess is that these properties co-occur with the names of the landmarks, events, and people in the training datasets and this is how they are learned by the LLM. It would be helpful if the authors can explain more about why they think that the LLM learning these properties is a sign of learning a world model.

W3. Only results from probing experiments are provided as evidence, and only when either the entity is given directly to the model as input or when a prompt that agrees with the task is appended to the context ("What is the location of.."). It would be informative to present the accuracy of the model under those prompts (e.g. is the probing needed to answer the question of whether the model has this information?), and also to show whether the representation of time and space is still decodable even if the task the model is asked to perform is not related to the time/location or is even adversarially related.

**Questions:**

Q1. Are the neurons that the authors point out as aligning with the space/time directions sufficient, necessary, or not even sufficient for representing these properties? In other words, what happens to the space/time representations if these neurons are ablated and the ridge regression probes are relearned?

Other questions: please respond to weaknesses 2 and 3 above.

Minor point:
page 3 is the first time when it's clear that what is meant by a spatial representation is the two-dimensional latitude and longitude coordinates. This should be made clear earlier in the paper.

---

> ### Author Response · Authors · 2023-11-20
>
> Thank you for your comments.
>
> > Q1. Are the neurons that the authors point out as aligning with the space/time directions sufficient, necessary, or not even sufficient for representing these properties?
>
> We do not believe these neurons are necessary or sufficient for representing these properties, but instead take them as strong evidence that the model does in fact learn the correct global representations. That is, because of how late these neurons occur in the model (typically around 50-70% depth), we believe that these neurons are “consumers” rather than “producers” of these representations, and use them for some downstream task.
>
> >In other words, what happens to the space/time representations if these neurons are ablated and the ridge regression probes are relearned?
>
> Given that the probe performance plateaus before this point, maximum probe predictive accuracy would not be affected.
>
> To better understand the function of these neurons, we perform new neuron ablation experiments, and analyze the LLM loss increase, rather than the difference in probe performance (see the general change log comment for analysis).
>
> > W2. It would be helpful if the authors can explain more about why they think that the LLM learning these properties is a sign of learning a world model.
>
> We think that having a coherent grasp of where entities are located in time or space is a necessary but not sufficient condition for having a world model in the strongest sense of the term.  That is, space and time and basic primitives that are required in a more comprehensive world model. We clarified this point in the introduction and discussion as we think our initial wording was not sufficiently clear.
>
> We also believe that these models are likely learning basic associations with co-occurrence statistics, at least initially, but that at some point these representations are organized into a representational space which actually reflects the external world, rather than simply clustering together co-occurring entities (as evidenced by the space/time neurons and hold-out generalization experiments which give evidence of global structure).
>
> > W3 It would be informative to present the accuracy of the model under those prompts
>
>  This is a good idea but we have found it very difficult to reliably prompt and parse the model for specific facts, due to the diversity of entity types within our dataset. This is made worse by the fact we are studying base models, and not chat models.
>
> > W3.b also to show whether the representation of time and space is still decodable even if the task the model is asked to perform is not related to the time/location or is even adversarially related.
>
> We actually do perform some experiments along these lines in Section 3.3 where we experiment with different prompts. For all datasets, we include an empty prompt which includes just the entity name (so no task specification) and find that performance is mostly preserved. We also experiment with adversarial prompts in the form of capitalizing the entity tokens, or prepend random tokens, and do observe a degradation in performance.
>
> > Minor point: page 3 is the first time when it's clear that what is meant by a spatial representation is the two-dimensional latitude and longitude coordinates. This should be made clear earlier in the paper.
>
> Thank you, we added a note of this in the third paragraph of the introduction.

---

> > ### Comment · Reviewer_eYoU · 2023-12-04
> >
> > Thanks for the response.
> > I'm still not quite clear on the claim of the model learning a world model. The authors say in their response:
> > "these representations are organized into a representational space which actually reflects the external world, rather than simply clustering together co-occurring entities (as evidenced by the space/time neurons and hold-out generalization experiments which give evidence of global structure)." -- as I understand it, the held out generalization is only showing that the space/time is available in the representations of the corresponding entity. However, it is not showing that the model can generalize to knowing the space and time of new entities that the model has not been trained on. To me, that is what a "world model" is: the ability to interpolate and extrapolate for new datapoints.
> > I am keeping my score as this was my main concern which was not addressed sufficiently. I suggest the authors really clarify in the updated manuscript.

---

### Official Review · Reviewer_xxLT · 2023-11-05

**Soundness:** 3 good
**Presentation:** 4 excellent
**Contribution:** 3 good
**Rating:** 8
**Confidence:** 4

**Summary:**

Do LLMs encode an enomous collection of superficial statistics, or do LLMs encode a coherent model of the data generation process (”a world model”)? This paper presents some evidence towards the latter.

This paper finds that Llama-2 family of models encode linear representations of space and time across multiple scales. Further, this paper finds “space neurons” and “time neurons” that reliably encode spatial and temporal coordinates.

**Strengths:**

- This paper presents important evidence towards a critical debate (what do LLMs encode).
- This paper organizes the probing studies in a way that is more comprehensive and rigorous than most probing papers that I have seen. To illustrate that the features are encoded linearly, this paper compares linear (ridge) regression probes and nonlinear MLP probes, and found that the nonlinear probes show minimal improvement in performance in any dataset or model. To illustrate the sensitivity to prompts, this paper tries many different types of prompts and discuss the effects. To test the robustness of the encoding, this paper sets up several block holdout and entity-holdout settings. These combinations of settings make the findings rigorous and compelling.
- The dimensionality reduction and space & time neuron experiments make this paper even more appealing.

**Weaknesses:**

The experiments only involve Llama-2 series models, whereas various locations in the paper stretches the claim to be about all LLMs and modern LLMs. I recommend rephrasing some texts into e.g., “LLMs, with Llama as examples” to make the claims better supported by the scope of the experiments.

Typo and comments:

- There are some minor typos. In page 6, a punctuation is needed before the footnote. In the end of the next paragraph, the period should be before the footnote.
- A related work should be referred: [Distributional vectors encode referential attributes](https://aclanthology.org/D15-1002) (Gupta et al., EMNLP 2015) This is one of the earliest probing papers, and it probed for the attributes involving geographic locations.

**Questions:**

- Do the findings generalize to other LLMs, for example bidirectional models?

---

> ### Author Response · Authors · 2023-11-20
>
> Thank you for your encouraging comments and helpful feedback!
>
> > The experiments only involve Llama-2 series models… Do the findings generalize to other LLMs
>
> Excellent point. To address this, we've now replicated our main results on the Pythia model suite to support the generality of our findings. We chose the Pythia family because it spans a large number of model sizes, has many training checkpoints, and is a popular model to study in the interpretability literature.
>
> As can be seen in the revised Figure 2, the main results continue to hold, with fairly clean scaling. However, because all Pythia models are only trained on 300B tokens, the large models are fairly under-trained compared to the 2T tokens used to train the LLama models.
>
> > There are some minor typos.
>
> Fixed. Thank you for your careful reading!
>
> > A related work should be referred
>
> Done: we added this reference to our new related work section on Linguistic Spatial Models.

---

> > ### Comment · Reviewer_xxLT · 2023-11-22
> > **Reviewer reply**
> >
> > Thank you for the response. The additional experiments indeed strengthen the main results. I'm happy to keep my original score.

---

### Author Response · Authors · 2023-11-20
**Comment to all reviewers regarding the revision.**

We thank the reviewers for their careful reading and thoughtful comments. We have uploaded a revised manuscript with changes marked in blue. A summary of the main changes
## Experiments
We reran our main probing experiment on 6 additional models, from the Pythia series (160M-6.9B), and updated Figure 2. The results show a very clear scaling trend, and emphasizes the importance of scale (a key aspect missing from prior work).

We also conduct several preliminary neuron ablation and intervention experiments. In the interest of getting the rest of our rebuttal out ASAP, our results are more qualitative and at the proof-of-concept phase, but we believe we have verified the basic importance of these neurons in spatial and temporal modeling. We intend to scale these experiments over the next few days to be more comprehensive.

These results are added to the appendix but in short:
We show that intervening on a single time-neuron can change the predicted decade of a song, movie, and book, much more than random neurons within the same layer.
We report the contexts with the maximum loss increase when zero-ablating a space or time neuron on a subset of wikipedia.

## Writing Revisions
The main changes concern clarifying our claim of finding a “literal world model.” We note there are multiple definitions for what a world model is, and intended to say “literal world model” here to have the most literal definition, i.e., a spatial model of the world. However, multiple reviewers have pointed out that "world model" is often taken to mean a model of the world that is not merely static, but includes causal dynamics. We have therefore reworded the manuscript to clarify that we're making only a weaker claim.  In particular, we emphasize that static spatiotemporal models are merely ingredients required in a more comprehensive dynamical world model, and that we show only that these ingredients exist, but not a full dynamical world model in the strongest sense of the term.

We have also clarified how our results suggest that the representations these models possess (and their internal use) are qualitatively different from representations found with much simpler methods raised in prior work, and that modern LLMs are reaching a scale where something more interesting is going on.

In line with this, we also added a paragraph to the related work section on “Linguistic Spatial Models” after reviewers raised additional older references we had missed. As emphasized in the individual rebuttals, we believe our work is far more comprehensive and at a much larger scale, with representations of hundreds of thousands of entities.

---

### Author Response · Authors · 2023-11-23
**Gentle reminder to respond**

As the end of the discussion period is approaching (November 22, EOD AOE), we would like to gently remind reviewers to see our rebuttals/revisions (marked in blue) and to please let us know if we have addressed their concerns or if they have any remaining questions.

---

> ### Comment · Reviewer_MD6k · 2023-11-23
> **Updated version is improved**
>
> After reading the updated paper, the authors' response and the other reviews I am revising my recommendation to 8. The revisions address the most serious concerns I had and I'd like to see the paper accepted.

---

### Meta-Review · Area_Chair_udLq · 2023-12-10

**Metareview:**

All reviewers agree this work provides a comprehensive analysis and interesting findings on how LLMs implicitly represent spatial coordinates and time across multiple scales. At the same time, they very consistently raised concerns about (i) similarity to previous works using word embeddings or more recent ones that make related claims about properties such as color or spatial directions.; (ii) misleading claims about world model representations and (iii) unclear generalization across different models and data. The author response was convincing, including new experiments to address (iii) and a revised write up to address (i) and (ii). However, while the authors did attempt to tone down the claims about learning world models, two reviewers remained concerned that the paper needs another revision to better address (ii). The AC recommends acceptance, as they consider this can be done with a minor revision, and believe the paper has interesting findings and a rigorous experimental analysis to support them, as acknowledged by all reviewers. We strongly encourage the authors to further clarify or remove any claims regarding learning world models in the paper — see detailed comments by reviewer XR65.

**Justification For Why Not Higher Score:**

Given the level of novelty of the paper (see detailed comments by the reviewers) and the fact that the current version still needs a minor revision to further tone down or remove the claims about learning world models, the AC considers the paper should not be given a higher score.

**Justification For Why Not Lower Score:**

The paper has interesting findings and a rigorous experimental analysis to support them, as acknowledged by all reviewers.

---

### Decision · Program_Chairs · 2024-01-16

Accept (poster)